# Interacting with Hemoglobin: *Paracoccidioides* spp. Recruits hsp30 on Its Cell Surface for Enhanced Ability to Use This Iron Source

**DOI:** 10.3390/jof7010021

**Published:** 2021-01-01

**Authors:** Aparecido Ferreira de Souza, Mariana Vieira Tomazett, Kleber Santiago Freitas e Silva, Juliana Santana de Curcio, Christie Ataides Pereira, Lilian Cristiane Baeza, Juliano Domiraci Paccez, Relber Aguiar Gonçales, Fernando Rodrigues, Maristela Pereira, Célia Maria de Almeida Soares

**Affiliations:** 1Laboratório de Biologia Molecular, Instituto de Ciências Biológicas, ICB II, Campus II, Universidade Federal de Goiás, Goiânia 74000-000, Brazil; aparecidofsouza@gmail.com (A.F.d.S.); mvtomazett@gmail.com (M.V.T.); smallbinho@hotmail.com (K.S.F.eS.); julianadecurcio1@gmail.com (J.S.d.C.); christierv@live.com (C.A.P.); julianopaccez@gmail.com (J.D.P.); maristelaufg@gmail.com (M.P.); 2Centro de Ciências Médicas e Farmacêuticas, Universidade Estadual do Oeste do Paraná, Cascavel 85819-110, Brazil; lilianbaeza@gmail.com; 3Life and Health Sciences Research Institute (ICVS), School of Medicine, University of Minho, 4700-000 Braga, Portugal; relbergoncales@usp.br (R.A.G.); frodrigues@med.uminho.pt (F.R.); 4ICVS/3B’s—PT Government Associate Laboratory, 4800-000 Guimarães, Portugal

**Keywords:** nutritional immunity, cell wall proteins, molecular dynamics

## Abstract

*Paracoccidioides* spp. are thermally dimorphic fungi that cause paracoccidioidomycosis and can affect both immunocompetent and immunocompromised individuals. The infection can lead to moderate or severe illness and death. *Paracoccidioides* spp. undergo micronutrients deprivation within the host, including iron. To overcome such cellular stress, this genus of fungi responds in multiple ways, such as the utilization of hemoglobin. A glycosylphosphatidylinositol (GPI)-anchored fungal receptor, Rbt5, has the primary role of acquiring the essential nutrient iron from hemoglobin. Conversely, it is not clear if additional proteins participate in the process of using hemoglobin by the fungus. Therefore, in order to investigate changes in the proteomic level of *P. lutzii* cell wall, we deprived the fungus of iron and then treated those cells with hemoglobin. Deprived iron cells were used as control. Next, we performed cell wall fractionation and the obtained proteins were submitted to nanoUPLC-MS^E^. Protein expression levels of the cell wall F1 fraction of cells exposed to hemoglobin were compared with the protein expression of the cell wall F1 fraction of iron-deprived cells. Our results showed that *P. lutzii* exposure to hemoglobin increased the level of adhesins expression by the fungus, according to the proteomic data. We confirmed that the exposure of the fungus to hemoglobin increased its ability to adhere to macrophages by flow cytometry. In addition, we found that HSP30 of *P. lutzii* is a novel hemoglobin-binding protein and a possible heme oxygenase. In order to investigate the importance of HSP30 in the *Paracoccidioides* genus, we developed a *Paracoccidioides brasiliensis* knockdown strain of HSP30 via *Agrobacterium tumefaciens*-mediated transformation and demonstrated that silencing this gene decreases the ability of *P. brasiliensis* to use hemoglobin as a nutrient source. Additional studies are needed to establish HSP30 as a virulence factor, which can support the development of new therapeutic and/or diagnostic approaches.

## 1. Introduction

Iron is one of the micronutrients that is highly essential for the metabolism of almost all organisms, but at high levels, iron is toxic and generates reactive oxygen species (ROS) [1]. Additionally, iron and other metals have their bioavailability strictly controlled by the mammalian host through a process called nutritional immunity, which affects the infectious success of pathogenic microbes [2]. Hence, microbial iron acquisition is an important virulence attribute [3]. Due to the importance of iron for the biological processes of both hosts and pathogens, but in contrast to the toxic potential of the metal, hosts employ the strategy of binding the metal to several proteins, such as ferritin, transferrin, lactoferrin and hemoglobin, in order to avoid both the metal toxicity and the access of pathogens to it [4].

When challenged by iron deprivation imposed by the host, pathogenic fungi employs mechanisms of high affinity and specificity to capture iron from the host, including reductive iron uptake pathways, biosynthesis, and uptake of siderophores, and the use of host’s iron-binding proteins [1,5,6]. Hemoglobin is an abundant host’s heme-containing protein and an important iron source for pathogens including fungi. The heme present in the hemoglobin can be removed and totally internalized by the fungi or the iron contained in the heme group can be released and readily internalized by pathogens [7,8]. *Candida albicans* captures hemoglobin-derived heme via a network of receptors that contain the CFEM domain (common in several fungal extracellular membrane proteins) that transfers the heme group to the ESCRT (endosomal sorting complex required for transport)-mediated endocytosis system [9,10,11,12,13]. *Cryptococcus neoformans* is also capable of using heme. The first step in the process is mediated by CIG1, a cell surface protein that, although not containing the CFEM domain, act as a hemophore [14]. Similarly to *C. albicans*, the process of heme endocytosis in *C. neoformans* is dependent on the ESCRT system [15,16]. In addition, recent work showed the importance of the clathrin-mediated endocytosis (CME) mechanism for the uptake of heme and hemoglobin in *C. neoformans*, given that mutants lacking this system are not successful in developing virulence factors. Interestingly, mutants *las17* (a gene encoding for a nucleation-promoting factor for actin assembly) were avirulent in mice infection tests [17].

The *Paracoccidioides* genus comprises thermally dimorphic fungi that grow in a hyphal form in the environment, but exist shift their morphology into budding yeast cells within mammalian hosts [18]. *Paracoccidioides* spp. infect at least 10 million people causing the disease paracoccidioidomycosis (PCM), which is geographically restricted to the subtropical areas of Latin America [19,20]. Members of the *Paracoccidioides* genus include the species *P. brasiliensis*, *P. lutzii*, *P. americana*, *P. restrepiensis,* and *P. venezuelensis*. The disease manifestations range from superficial skin lesions to invasive infections [21,22]. The infection is initiated by inhalation of fungal propagules, which differentiate into yeast cells after reaching the alveolar epithelium [23].

When challenged with iron deprivation, *Paracoccidioides* spp. employs a complex response that encompasses transcriptional and metabolic reprogramming and the expression of high affinity systems for iron uptake, including biosynthesis and uptake of siderophores and a non-canonical iron reductive pathway [24,25,26]. Additionally, we have demonstrated previously that hemoglobin is the preferential host iron source of the *Paracoccidioides* spp. These fungi have hemolytic activity, able to acquire iron from hemoglobin and endocytize the heme group without releasing iron into the extracellular level. A GPI-anchored receptor, Rbt5, has the primary role of acquiring the essential nutrient iron from the host hemoglobin [27].

Previous studies developed at our laboratory allowed the identification of proteins from *P. lutzii* cell wall by using subcellular fractionation and nanoUPLC-MS^E^. Sequential fractionation associated with a proteomic approach allowed the characterization of *P. lutzii* cell wall proteins, such as proteins associated with the cell wall through non-covalent or disulfide bonds (F1) and proteins bound to the cell wall by alkali-labile bonds (F2) [28]. We identified proteins present in the cell wall that belong to several functional categories. Some of these proteins have been previously characterized in the cell wall of *Paracoccidioides* genus playing putative roles in cell wall biogenesis and organization as well as in virulence, adhesion, colonization, survival in hostile environments and evasion of the immune system, thereby allowing the establishment of the disease [29,30,31,32,33]. Another study also carried out in our laboratory allowed the identification of *P. lutzii* cell surface proteins that interact with macrophages through affinity chromatography based on surface proteomics [34]. The authors obtained F1 cell wall enriched extracts and identified proteins, such as serine proteinase and fructose-1,6-bisphosphate aldolase interacting with macrophage surface proteins, and confirmatory assays showed that these proteins undergo positive regulation during the interaction of the fungus with the host [34]. The studies performed by Araújo et al. (2017) [28] and Tomazett et al. (2019) [34] demonstrated that the identification of the cell wall proteins is a good alternative to list new targets in the context of host–pathogen interaction.

Therefore, in order to investigate changes in the proteomic level of *P. lutzii* cell wall, we deprived the fungus of iron and then treated those cells with hemoglobin. Deprived iron cells were used as control. Next, we performed cell wall fractionation and the obtained proteins were submitted to nanoUPLC-MS^E^. Protein expression levels of the cell wall F1 fraction of cells exposed to hemoglobin were compared with the protein expression of the cell wall F1 fraction of iron-deprived cells. Our results demonstrated that upregulation of potential adhesins occurs at the cell surface when the fungus is exposed to hemoglobin and revealed that HSP30 of *P. lutzii* is a novel hemoglobin-binding protein. In order to investigate the importance of *Pb*HSP30 in the *Paracoccidioides* genus, we developed a *P. brasiliensis* (*Pb*18) knockdown strain of HSP30 via *Agrobacterium tumefaciens*-mediated transformation (ATMT) and demonstrated that silencing this gene decreases the ability of *P. brasiliensis* to use hemoglobin as a nutrient source. These findings highlight the importance of HSP30 regarding the use of hemoglobin by *Paracoccidioides* spp. and we hypothesize that HSP30 may be important in the context of infection, given that the use of hemoglobin by the fungus is a virulence attribute. To confirm this inference, further investigation is needed. Certainly, these studies will contribute to the expansion of the knowledge regarding *Paracoccidioides*–host interaction, which can culminate in the development of new therapeutic and/or diagnostic approaches.

## 2. Materials and Methods

### 2.1. Ethics Statement

All experiments were conducted in accordance with the Brazilian Federal Law 11,794/2008 establishing procedures for the scientific use of animals. All efforts were made to minimize suffering, and the animal experiments were approved by the Ethics Committee on the use of Animal Experimentation (Federal University of Goiás, CEUA-UFG; under protocol number 116/17, approved on 13 November 2017), following the guidelines of the National Council for Control of Animal Experimentation.

### 2.2. Strains and Growth Conditions

*P. lutzii* (American Type Culture Collection—ATCC MYA-826—*Pb*01) and *P. brasiliensis* (ATCC 32069—*Pb*18) were maintained in Brain Heart Infusion (BHI) solid broth supplemented with glucose 4% (*w/v*) at 36 °C. To obtain cell wall proteins extracts, *P. lutzii* yeast cells were cultured in BHI liquid medium, supplemented with glucose 4% (*w/v*) for 72 h at 36 °C and under shaking at 120 rpm. Next, 5 × 10^6^ cells/mL were transferred to McVeigh and Morton modified medium (MMcM) chemically defined liquid medium [35] without iron, supplemented with 50 µM of the iron chelator bathophenanthrolinedisulfonic acid (BPS—Sigma-Aldrich, St. Louis, MO, USA) and were maintained in this condition for 36 h in order to establish intracellular iron depletion. Finally, yeasts cells were transferred to MMcM containing bovine hemoglobin 10 µM (Sigma-Aldrich, St. Louis, MO, USA) or BPS 50 µM and maintained for 48 h.

### 2.3. Extraction of P. Lutzii Cell Wall Proteins (CWPs)

To obtain *P. lutzii* CWPs, we employed the methodology described by Araújo et al., (2017) [28], with some modifications. After 48 h of fungal culture in the presence of hemoglobin (treatment) or in iron deprivation (control), the cultures were collected and centrifuged at 800× *g* for 10 min at 4 °C, and cells were washed 5 times with lysis buffer (10 mM Tris-HCl pH 8.5, 2 mM CaCl_2_, 1:10 protease inhibitor). Subsequently, the cells were resuspended in ice-cold 10 mM Tris-HCl buffer, pH 8.0, broken in the presence of liquid nitrogen and the cellular powder was resuspended in ice-cold lysis buffer. Then, glass beads (4 mm) were added to the suspension. The mixture was vortexed for 1 h at 4 °C. After centrifugation at 800× *g* for 10 min at 4 °C, the glass beads were removed and pellets were washed 5 times with ice-cold ultrapure water, 5 times with 0.86 M NaCl, 5 times with 0.34 M NaCl and 5 times with 0.17 M NaCl. The washes performed with NaCl solutions aimed to remove potential contaminants, which can range from extracellular proteins to membrane or cytoplasmic proteins that can be retained on the isolated cell wall by non-specific ionic interactions [36]. The obtained pellets were treated twice with extraction buffer (50 mM Tris-HCl, pH 7.8, 69 mM SDS, 10 mM EDTA and 40 mM Mercaptoethanol) for 10 min at 100 °C and then centrifuged. The supernatants were collected, concentrated using membranes with 10-kDa exclusion level (Amicon Ultra centrifugal filter, Millipore, Bedford, MA, USA) and washed 3 times with ice-cold ultrapure water. Afterwards, treatment with 2-D Clean-Up Kit (GE Healthcare) was performed with another three washes with ice-cold ultrapure water. The obtained extract corresponded to proteins that are non-covalently linked or linked to the cell wall by disulfide bonds.

### 2.4. Preparation of Complex Samples for NanoUPLC-MS^E^

The F1 CWPs extracts were quantified. We added 150 µL of a 0.2 % (*w/v*) RapiGest solution and 10 µL of 50 mM NH_4_HCO_3_ at pH 8.5 to 300 µg of protein extracts. The mixture was incubated at 80 °C for 15 min and subsequently, 2.5 µL of 100 mM DTT were added, followed by another incubation at 60 °C for 30 min. Next, 2.5 µL of 300 mM of iodoacetamide were added and the samples were maintained for 30 min away from the light at room temperature. Samples were then subjected to tryptic digestion with 60 µL of a 0.05 µg/µL trypsin solution and incubated at 37 °C for 16 h. Then, 60 µL of 5% (*v/v*) trifluoroacetic acid were added and the solution was incubated at 37 °C for additional 90 min. The samples were centrifuged at 13,000× *g* for 30 min at 4 °C and the supernatants were transferred to clean tubes. The centrifugation process was repeated until there was no further precipitate formation. The samples were concentrated in a speed vacuum apparatus, resuspended in 12 µL of ultrapure water, purified on ZipTip Pipette Tips (ZipTip C18 Pipette Tips, Millipore, Bedford, MA, USA) following the manufacturer’s instructions, and again concentrated in a speed vacuum concentrator. The obtained peptides were resuspended in 60 µL of 20 mM ammonium formiate, at pH 10, containing 200 fmol/µL phosphorylase B (PHB) (Waters Corporation, Manchester, UK) (MassPREPTM protein), used as the internal control. The obtained samples were submitted to nanoUPLC-MS^E^, as described in the subsequent session.

### 2.5. Data Acquisition by NanoUPLC-MS^E^

The samples were subjected to a high-resolution liquid chromatography at the nanoscale through the ACQUITY UPLC^®^ M-Class system (Waters Corporation, Manchester, UK). The fractionation of the peptides was performed in an XBridge^®^ Peptide 5 µm BEH130 C18, 300 µm × 50 mm reverse phase pre-column (Waters Corporation, Manchester, UK). The first dimension of the system was maintained in a flow rate of 0.5 µL/min with an initial condition of 3% acetonitrile (ACN). The peptides were submitted to five fractions (F1–F5) of an ACN gradient (F1–11.4%, F2–14.7%, F3–17.4%, F4–20.7%, and F5–50%). For the second dimension, each fraction was eluted on a 2D Symmetry^®^ 5 µm BEH100 C18, 180 µm × 20 mm trapping column (Waters Corporation, Manchester, UK) and passed through the analytical column Peptide CSH^TM.^BEH130 C18 1.7 µm, 100 µm × 100 mm (Waters Corporation, Manchester, UK), in a flow of 0.4 µL/min at 40 °C. The human (Glu1)-fibrinopeptide B (GFB-Sigma-Aldrich, St. Louis, MO, USA) protein was used for the mass calibration, which was measured every 30 s and in a constant flow of 0.5 µL/min. GFB was used at the concentration of 200 fmol/µL. Peptide identification and quantification were carried out in a Synapt G1 MSTM (Waters Corporation, Machester, UK) mass spectrometer equipped with a NanoElectronSpray source and two mass analyzers [a first quadrupole and the second time of flight (TOF). Three experimental replicates were performed.

### 2.6. Spectra Processing and Proteomic Analysis

After nanoUPLC-MS^E^, data processing was performed using ProteinLynx Global Server version 3.0.2 software (PLGS) (Waters Corporation, Manchester, UK), which allowed the determination of the exact mass retention time (EMRT) of the peptides and their molecular weight, by mass/charge ratio (m/z). For identification of the peptides, the obtained spectra (along with reverse sequences) were compared with sequences available in the database of *P. lutzii* (*Pb*01) (https://www.uniprot.org/proteomes/UP000002059). To refine protein identification, we employed the detection of at least two ions per peptide fragments, five by protein fragments, the determination of at least one peptide per protein, false positive rate of 4%, carbamidomethylation of cysteine, oxidation of methionine, phosphorylation of serine, threonine and tyrosine and a trypsin-lost cleavage site was allowed. The tolerable mass error for identification of the peptides was set to 50 ppm. The quality graphics for the races were generated through MassPivot software v1.0.1 (kindly provided by Dr. André Murad), FBAT software (https://sites.google.com/view/fbat-web-page), Spotfire^®^ v8.0 (TIBCO Software Inc.^©^, Palo Alto, CA, EUA) and Microsoft Office Excel (Microsoft^®^, Redmond, Washington, DC, USA). Proteins present in at least two of the three experimental replicates of extracts were included in the subsequent differential expression analysis. Proteins that presented the lowest coefficient of variance and that were detected in all the replicates were used for the normalization of intensity and the Expression Algorithm (Expression^E^, Waters Corporation, Manchester, UK ), which is part of the PLGS software [37], was used for the differential expression analysis. Proteins classified as regulated presented fold change ± 0.5 between the quantification of the extract obtained in the presence of hemoglobin × Fe deprivation. Homology investigation of identified hypothetical proteins was performed using the BLASTp online tool (Basic Local Alignment Search Tool—https://blast.ncbi.nlm.nih.gov/Blast.cgi?PAGE=Proteins). Protein sequences were further subjected to in silico analysis for signal peptide prediction using the Signal P 4.0 Server online tool (http://www.cbs.dtu.dk/services/SignalP-4.0/). For the prediction of proteins secreted by non-classical pathways, the online tool Secretome P 2.0 was used. The prediction of potential adhesins was performed using the online FaaPred tool (http://bioinfo.icgeb.res.in/faap/). Heatmap graphic was generated by Microsoft Office Excel (Microsoft^®^, Redmond, Washington, DC, USA). The HSP30 protein was an important finding in our proteomic data and we decided to investigate this result further.

### 2.7. Expression of the Recombinant HSP30 Protein in Escherichia Coli, Protein Purification, and Polyclonal Antibodies

Total RNA was extracted from fungal yeast cells using the TRIzol reagent (TRI Reagent^®^, Sigma-Aldrich, St. Louis, MO, USA) and mechanical cell rupture (MiniBeadbeater—BioSpec Products), as described by the manufacturer’s protocol. From the extracted RNA, cDNA was synthetized following the manufacturer recommendation of the Super Script^®^ Reverse Transcriptase Kit (Invitrogen™, Waltham, MA, USA). The cDNA was used to amplify the HSP30 protein (PAAG_00871) using the polymerase High Fidelity (Invitrogen™, Waltham, MA, USA). The oligonucleotides sense sequence was 5′GGTTCCGCGTGGATCCATGTTCTCTCGTCGAGCC′3 and the antisense was 5′GGGAATTCGGGGATCCCTACTCAATCGTAATCTTCTT′3. The amplification cycle comprised denaturation at 94 °C for 2 min followed by 40 cycles of denaturation at 94 °C for 30 s, annealing at 54 °C for 30 s and extension at 72 °C for 1 min and 30 s, followed by a final extension at 72 °C for 5 min. The cDNA product obtained by RT-PCR was cloned into the expression vector pET-32a. Bacterial cells, strain *Escherichia coli* C43, harboring the recombinant plasmid were grown in Luria-Bertani (LB) medium supplemented with 100 µg/mL ampicillin (*w/v*) under agitation at 37 °C until the optical density (OD) reached an absorbance of 0.6 at a wavelength of 600 nm. The reagent Isopropyl-β-d-thiogalactopyranoside (IPTG) was added to the growing culture to a final concentration of 0.1 mM. The bacterial cells were harvested by centrifugation at 10,000× *g* for 10 min after 16 h of incubation at 15 °C and resuspended in phosphate buffered saline (PBS) 1X. The recombinant HSP30 protein fused to Trx-His-Tag was used for the production of polyclonal antibodies in 4 BALB/c male mice aged 6–8 weeks. The fusion protein was removed from SDS-PAGE gels and injected into mice along with Freund’s adjuvant three times at intervals of 15 days. Serum containing polyclonal antibodies was collected and stored at −20 °C. The protein was produced in inclusion bodies and was solubilized using 50 µL of a 20% (*w/v*) *N*-Lauroylsarcosine sodium salt (Sigma Aldrich, Missouri, KS, USA) solution for 5 mL of bacterial extracts and sonicated (5 times, 10 min). SDS-PAGE analysis showed the protein in the soluble fraction and then, the protein was purified by a nickel resin chromatography system (Qiagen Inc., Germantown, MD, USA).

### 2.8. Far-Western Blot Analyses

We used the protocol described by [38] to perform far-western blot analysis. The recombinant HSP30 protein and bovine hemoglobin were transferred to Hybond nitrocellulose membranes (GE Healthcare, Piscataway, NJ, USA) and incubated for 1 h at room temperature with bovine hemoglobin diluted at 35 µg/mL in blocking buffer containing 10% (*w/v*) skim milk powder and 0.1% (*v/v*) Tween-20 in PBS 1X. Washing of the membrane was performed, followed by incubation with the primary monoclonal antibody anti-human hemoglobin produced in mice (Abcam Plc, Cambridge, UK) and diluted 1:100 in blocking buffer. Another washing series was performed and the membrane was incubated with anti-mouse alkaline phosphatase conjugated secondary antibody (1:10,000) for 1 h at room temperature and protected from light. The revelation step was performed employing 5-bromo-4-chloro-3-indolyl phosphate/p-nitroblue tetrazolium chloride (BCIP/NBT).

### 2.9. Flow Cytometry and Immunofluorescence Assays

*P. lutzii* yeast cells were cultivated in the presence of 10 µM of hemoglobin or 50 µM of BPS. Two additional conditions were included, which were 10 µM of FeSO_4_ and 10 µM of Bovine Serum Albumin (BSA). Flow cytometry was performed according to the methodology described by [34]. Murine macrophages of the lineage RAW 264.7 were cultured in RPMI medium containing bovine fetal serum 10% (*v/v*) and MEM non-essential amino acid solution (Sigma Aldrich, Missouri, KS, USA) at 36 °C and 5% CO_2_ until complete confluence. The experiment was performed in 12-well polypropylene plates (Greinner Bio-One, Monroe, NC, USA). We plated 10^5^ macrophages per well in Roswell Park Memorial Institute (RPMI) medium containing gamma interferon (IFN-γ) (1 U/mL) (Sigma-Aldrich, St. Louis, MO, USA) following incubation for 24 h at 36 °C and 5% CO_2_ for adherence and activation. Then, the medium was replaced by a fresh RPMI medium containing IFN-γ (1 U/mL) and 5 × 10^5^
*P. lutzii* yeast cells per well were added to the macrophages. The cells were incubated for 4 h at 36 °C and 5% CO_2_. Macrophages were washed with PBS 1X to remove unbound fungal cells, fixed with cold methanol for 2 h at −80 °C and the cells were collected by scraping. The cells were washed three times with PBS 1X, incubated with 100 µg/mL Congo Red (Sigma-Aldrich, St. Louis, MO, USA) for 15 min and washed 3 times again with PBS 1X. The number of yeast cells of *P. lutzii* adhered/internalized to macrophages was identified by flow cytometry assay and the instrument Guava^®^ easyCyte (Merck Millipore, Darmstadt, Germany) acquired a minimum of 10,000 cells per sample. Proportion test was used for statistical comparison between the conditions analyzed [39].

For the immunofluorescence assays, *P. lutzii* yeast cells were grown in the absence or presence of hemoglobin at 10 µM for 48 h at 36 °C under agitation. Subsequently, the cells were centrifuged, and the pellet washed with PBS 1X. Cells were counted in a Neubauer’s chamber and 10^6^ cells/mL were fixed with ice-cold methanol for 2 h at −80 °C. Then, cells were washed three times with PBS 1X, blocked for 30 min with PBS 1X containing BSA (3% (*w/v*)) and tween-20 (0.2% [*v/v*]) at room temperature and washed 3 times again with PBS 1X. The cells were incubated for 1 h at room temperature with anti-HSP30 protein polyclonal antibodies, produced in mice and diluted 1:100 in blocking buffer. Following 3 washes with PBS 1X, the cells were incubated for 1 h at room temperature with anti-mouse IgG coupled with fluorescein isothiocyanate (FITC- Sigma-Aldrich, St. Louis, MO, USA) diluted 1:100 in blocking buffer. The cells were washed with PBS 1X 3 times and observed under an Axio Scope A1 fluorescence microscope at bright field and 470/440 nm wavelength. Digital images were acquired using the software AxioVision (Carl Zeiss AG, Berlin, Germany).

### 2.10. Structural Alignment of HSP30 and Human Heme Oxygenase

The sequence alignment of HSP30 (PAAG_00871) and human heme oxygenase 1 (P09601) was obtained by the ClustalX 2.1 program [40]. In order to perform protein structure comparisons we used the TM-align algorithm [41]. This approach was based on the generation of an enhanced residue-to-residue alignment through dynamic simulations considering the backbone alpha-carbon coordinates of the HSP30 and hemoglobin. The best model was then produced via superposition of the structures under study and a TM-score was generated. The sequence identity cutoff was set to 95% and gap penalties were eliminated to circumvent fragmentation caused by the topological complexity of helices. The PDB output was used in PyMOL (pymol.org) to compare the alignment results from the TM-align server.

### 2.11. Preparation of Three-Dimensional (3D) Structures and Molecular Docking

The Iterative Threading Assembly Refinement (I-TASSER) server [42] was used to model the three-dimensional structure of HSP30 from *P. lutzii*. The server models protein structures based on templates of homologous proteins experimentally determined and deposited on the PDB (protein data bank) database. I-TASSER applies fold recognition via Monte Carlo simulations in order to rank homologous fragments and the procedure follows basic steps consisting of prediction of secondary structure by Protein Secondary Structure Prediction (PSSpred) and identification of threading templates by Local Meta-Threading-Server (LOMETS). Fragments are assembled in clusters according to conformation and energy levels to detect native similar structures. Finally, the modeled structure undergoes molecular dynamics refinement and prediction of biological function by COACH. The quality of the HSP30 three-dimensional structure was assessed through the MolProbity server.

The *Homo sapiens* hemoglobin structure was retrieved from the PDB under the accession number 1A3N. The ClusPro protein–protein anchor server [43] was used to determine the best protein complex conformations between HSP30 and hemoglobin. After the assembly of the complexes, the next step was to identify the amino acids involved in the interaction. We used the KFC2 server [44] to recognize all residues from the interaction interface from the complex formed by HSP30 and hemoglobin.

### 2.12. Molecular Dynamics Simulations

Molecular dynamics simulations determined a stable structure of the complex, similar to the native structure model. The protein complex under study was submitted to molecular dynamics using the software GROMACS 4.5.5, AMBER force field (ff99SB-ILDM) with the presence of explicit water TIP3P solvent [45]. The first step of the simulation is the minimization of the overall free energy to remove unfavorable contacts, carried on until 1000 kJ/mol or until the system reached the number of pre-determined steps. We performed a 100 ps NVT (mol, volume, and temperature) simulation followed by a 100 ps NPT (mol, pressure, and temperature) simulation in order to guarantee the balance of the thermodynamic variables. The NVT simulation allowed the pressure of the system to vary and at this step, the temperature was set to 300 K and velocities were calculated through Maxwell’s equations. During the NPT simulation, the volume is variable and the pressure was maintained by the Parrinello–Rahman barostat.

After the preparation of the system was completed, the complex formed by HSP30 and hemoglobin was submitted to a simulation of 200 ns, 300 K, 1 atm and time interval of 2 fs. No conformation restriction was applied at this point. Molecular dynamics analysis of trajectories was performed by root mean square deviation (RMSD) in relation to the initial structure by the *gromos* algorithm [46]. The g_cluster program (GROMACS package) was used to determine the most frequent conformations of the structure during the simulation. We used a cut-off of 0.6 nm to distinguish the conformational sets based on the RMSD profile. The cluster analysis and the RMSD allowed the analysis of the protein profile throughout the simulation and the most representative conformational mode was selected to undergo further analysis. Before the calculation of RMSD evolution and before analyzing clusters form the simulation trajectories, the structures were centered and fitted accordingly.

### 2.13. Construction of P. Brasiliensis HSP30 Antisense (AsHSP30) Strain

To obtain the silenced strain for the *P. brasiliensis HSP30* gene, the ATMT methodology was used as previously described [27,47,48]. DNA of *P. brasiliensis* was obtained after culturing the cells in BHI medium at 36 °C for 72 h at 150 rpm. The oligonucleotides used to amplify the sequence corresponding to aRNA of *Pb*HSP30 were HSP30 forward 5′ CTCGAGCGGGCTCCAAAGA 3′ and HSP30 reverse 5’ GGCGCGCCGGATGCTCAT 3′. The amplified fragment was inserted into the pCR35 plasmid under the control of the promoter region of the calcium-binding protein gene (CBP-1) from *Histoplasma capsulatum* [49]. Next, the CBP-1 promoter-AS cassette was subcloned into the pUR5750 plasmid, harboring a hygromycin B phosphotransferase as a selection mark. We used this system as a parental binary vector to harbor the aRNA cassette within the transfer DNA (T-DNA) [50]. These constructions were introduced into *A. tumefaciens* LBA1100 strains by electroporation and isolated via kanamycin selection (100 mg/mL). *P. brasiliensis* yeast cells were co-cultivated with transformed *A. tumefaciens* for 3 days at 25 °C and the selection of *P. brasiliensis* transformed cells was performed in BHI solid media containing hygromycin B (75 mg/mL) after 15 days of incubation at 36 °C. For the mitotic stability, randomly selected colonies of *P. brasiliensis* were grown for nine additional cycles in BHI solid medium, alternating absence and presence of 75 mg/mL of hygromycin.

### 2.14. Characterization of the Knockdown Strain

The HSP30 knockdown strain was characterized according to the level of transcript, growth in different media and cell viability analysis. For the transcriptional analyzes, wild tipe (WT) and silenced HSP30 strains were cultured in MMcM containing hemoglobin at 10 µM. After RNA extraction with TRIzol, the samples were treated with DNAse (RQ1 RNase-free DNase, Promega) and subjected to in vitro reverse transcription (SuperScript III First-Strand Synthesis SuperMix; Invitrogen, Life Technologies), using Oligo(dT). cDNAs were used in the RT-qPCR reaction through the QuantStudio Real-Time PCR (Thermo Fisher Scientific) with a mixture of SYBR green PCR master mix (Applied Biosystems, Foster City, CA, USA). The normalizing gene was the transcript encoding to glyceraldehyde-3-phosphate dehydrogenase (GenBank XM_015846519.1) as determined by Norm-Finder test [51]. The oligonucleotides are depicted in Appendix A. Cultures were grown in BHI medium and cellular density was measured in triplicates in a spectrophotometer Ultrospec 2000 (Pharmacia Biotech, Piscataway, NJ, USA). The viability of strains was determined by propidium iodide staining and detected by fluorescence microscopy (493/636 nm) in an Axioscope A1 microscope (Carl Zeiss AG, Berlin, Germany) [48]. To evaluate the ability of silenced strains to use hemoglobin, we subjected them to the above described culture condition and plated in triplicate in solid BHI medium. Colony forming units (CFU) were determined by counting. Statistical analysis was performed by the Student’s T test and *p* values ≤ 0.05 were considered statistically significant.

## 3. Results

### 3.1. Hemoglobin Promotes Changes at the P. Lutzii Cell Wall Proteome

*P. lutzii* yeast cells were incubated with hemoglobin or BPS (control) and a specific protocol for the isolation of cell wall proteins was employed, as described in the MM section. Protein extracts were submitted to nanoUPLC-MS^E^ and the proteomic data were analyzed regarding experimental quality standards that validated the analysis, as shown in Appendix A [37]. We found 83 proteins upregulated when yeast cells were in the presence of hemoglobin, as shown in Appendix A. To refine the results, bioinformatics tools were used to analyze which of the identified proteins had predicted secretion signal sequences, i.e., that potentially were addressed to the cell surface. This analysis revealed 33 proteins predicted to reach the fungal cell surface, representing 39.8% of the identified proteins.

For subsequent analysis, we selected proteins with positive prediction of secretion and with increased expression in the presence of hemoglobin. Table 1 and Figure 1A summarize the 33 proteins with predicted secretion. Aspartate-tRNA (Asn) ligase (PAAG_05117; XP_015699700) showed the highest fold change (5.53). Six proteins were identified exclusively in the presence of hemoglobin: HSP30 (PAAG_00871; XP_002797012.2), 40S ribosomal protein S17 (PAAG_01413; XP_002796405.1), 40S ribosomal protein S26E (PAAG_07847; XP_002790158.1), membrane-associated progesterone receptor component 1 (PAAG_01861; XP_002795973.1), protein disulfide-isomerase domain (PAAG_11504; XP_015703276.1) and pyruvate dehydrogenase protein X component (PAAG_02769; XP_002795293.1). Figure 1A presents the heatmap of upregulated proteins predicted as secreted in the presence of hemoglobin.

In order to correlate protein and transcript levels, firstly, we analyzed the transcript expression of HSP30 (PAAG_00871). The results showed that within 24 h, *P. brasiliensis* expresses approximately 20% more HSP30 transcripts in the presence of hemoglobin when compared to the control (BPS) (Figure 1B), but there was a decrease in the level of its expression after 48 h. Hence, mRNA levels increase after 24 h of treatment before protein levels increase.

Next, we determined which upregulated proteins were potential adhesins (Table 1) using bioinformatics tools. Our search revealed 7 potential adhesins and three of them were identified solely in yeast cells submitted to hemoglobin treatment: disulfide-isomerase domain protein (PAAG_11504; XP_015703276.1), pyruvate dehydrogenase protein X component (PAAG_02769; XP_002795293.1) and HSP30 (PAAG_00871; XP_002797012.2). In order to confirm the increase in the adhesion capacity of *P. lutzii*, we pretreated yeast cells with hemoglobin and assessed the ability of cells to interact with macrophages by flow cytometry, as shown in Figure 2. Statistical analysis using proportion test showed that the three conditions analyzed showed a significant difference when compared to the control condition (BSA). However, using the same test, the difference in the ratio between BSA and hemoglobin was 0.54. For BPS and FeSO_4_, conditions also tested, this difference was 0.11 and 0.02, respectively. Thus, a comparison between the conditions showed that the treatment of *P. lutzii* cells with hemoglobin increased the interaction with macrophages by 5 and 25 times when compared with BPS and FeSO_4_, respectively. This finding demonstrates that hemoglobin strongly modulates the ability of *P. lutzii* to adhere to macrophages.

### 3.2. Hemoglobin Increases HSP30 Expression at Cell Surface

The treatment of yeast cells with hemoglobin increases the fungal cell adhesion capacity to macrophages. The 30-kDa protein, present at the fungus F1 cell wall fraction, showed potential adhesin properties. The recombinant protein was produced in bacteria, purified, confirmed by spectrometry, and used for polyclonal antibodies production in mice, as depicted in Appendix A. Yeast cells were employed for immunofluorescence analysis, as depicted in Figure 3. Fluorescence of HSP30 and its recognition by antibodies at the fungal cell surface increased when the fungus was previously cultivated in the presence of hemoglobin, which corroborates our proteomic data. No reaction was obtained by using the non-immune sera, as expected.

### 3.3. HSP30 Binds to Hemoglobin at the Fungus Cell Wall by Hydrogen Bonds

As previously shown, HSP30 expression increased when *P. lutzii* cells were grown in the presence of hemoglobin. This finding led us to question whether these proteins interact directly. Hence, we performed analyzes of molecular dynamics. At first, the simulation performed on the complex formed by HSP30 and hemoglobin showed that the evolution of the system achieved a better conformation and provided insight into the molecular motion of the complex on an atomic scale. Before molecular dynamics, the system presented 85 unfavored rotamers and after the simulation, this number reduced 83%. Similarly, the number of favored rotamers before and after the simulation was improved by 200% (Appendix A).

Molecular dynamics allowed the trajectories of atoms and bonds in the HSP30-hemoglobin complex to be determined numerically through Newton’s equations of motion. The forces between the proteins and their potential energies were calculated based on interatomic potentials and force field (AMBER). We determined 103 poor quality bonds within the complex before the simulation and this number fell to zero after the molecular dynamics, showing the best quality of the resultant complex conformation. In addition, there were 11 twisted peptides identified in the complex before the simulation and five after the simulation.

Ramachandran plots were used to analyze energetically allowed regions for backbone angles and amino acid residues in the complex structure. The number of amino acid residues in energetically favored regions increased from 82.7% to 90.4% after the molecular dynamics; regarding allowed regions, the number of residues increased from 95.8% to 98.8% (Appendix A). This indicates a stereochemical improvement of the three-dimensional model of the complex.

The quality of the complex formed by the interaction between HSP30 and hemoglobin was assessed by RMSD and cluster analysis after the molecular dynamics simulation. The analyses of the trajectories allowed identifying the equilibration period, the quality of the simulation and the clusters with similar conformations. The simulations became more stable for RMSD at a value of 0.5 nm regarding the complex formed by the interaction between HSP30 and hemoglobin (Appendix A). Analyzed individually, the simulation of hemoglobin achieved stability around and RMSD value of 0.6 nm (Appendix A) and HSP30 achieved stability around 1.2 nm (Appendix A). These results show that the conformational stability of the complex is higher compared to the structures of the proteins when they are not interacting. According to the trajectories of the simulations, we identified 18 conformational clusters. Cluster analysis showed that the complex conformation of the first three clusters are more stable. In addition, the first cluster remained stable for a longer period than the other clusters, persisting throughout the simulation of the trajectory (Appendix A).

We predicted the interaction between HSP30 and hemoglobin in silico. The interface of interaction between the proteins is large (Figure 4) and maintained by several amino acid residues that interact via hydrogen and polar bonds. Figure 4A shows the surface of the complex and the interaction interface between HSP30 and hemoglobin. Figure 4B shows the cartoon view of the complex and the secondary structures of each protein involved in the interaction. The most important residues that contribute to the free energy of the complex are highlighted in Figure 4 and they interact through hydrogen bonds ranging from 1.5 to 2.8 Å. Asparagine (N80) of hemoglobin interacts with glutamic acid (E232) of HSP30 via a 2.5 Å hydrogen bond (Figure 4C). Arginine (R199) of hemoglobin interacts with phenylalanine (F45) of HSP30 distant 2.4 Å (Figure 4D). The residue F46 of hemoglobin performs two hydrogen bonds with tyrosine (Y222) and E230 of HSP30, distant 2.8 and 2.4 Å, respectively. In addition, the amino acid residue E230 interacts with another residue from hemoglobin, R40, through a hydrogen bond of 1.8 Å (Figure 4E). Finally, H2 (histidine) and Y545 of hemoglobin interacts with V338 and S339, via hydrogen bonds distant 2.5 and 1.7 Å, respectively (Figure 4F). Additionally, we investigated if the interaction of these two molecules would occur in vitro. Through far-western analysis, we demonstrate that HSP30 can bind to hemoglobin, reinforcing the role of this molecule as a potential adhesin, according to the in silico analysis (Figure 4G).

### 3.4. Knockdown of HSP30 Promotes Decreased Cell Growth Post Cultivation in Medium Containing Hemoglobin as Sole Iron Source

To generate *AsHSP30 P. brasiliensis* yeast cells, we used the ATMT methodology. The yeast cells of the *P. brasiliensis* wild type strain were transformed with T-DNA containing *AsHSP30* (Figure 5A). The silenced strain grown in BHI medium showed similar growth and viability levels to the wild type (Figure 5B). However when in the presence of hemoglobin the HSP30 silenced strain showed a decrease in mRNA levels, compared to the wild strain (Figure 5C). In addition, in order to evaluate if the silenced strain presented any defect in the ability to use hemoglobin, we deprived such cells of iron, exposed them to hemoglobin, and evaluated their growth by colony forming units (CFU) counting in BHI medium. The silenced strain had a lower number of CFU when compared to the wild type (Figure 5D). This finding reiterates the role of HSP30 in the use of hemoglobin by *Paracoccidioides*.

## 4. Discussion

The host–pathogen interaction is remarkably complex and molecular mechanisms on both sides promote a real competition for survival. The host’s nutritional immunity and the mechanisms that pathogens use to overcome it, occupy a prominent place in this context, as demonstrated for several pathogenic bacteria and fungi [5,7]. Previous works by our group have shown that fungi of the *Paracoccidioides* genus employ different strategies to counteract iron limitation, such as the remodeling of the metabolism prioritizing non-iron dependent pathways for energy production, biosynthesis, secretion and uptake of siderophores and the reductive pathway of iron assimilation [24,25,26,48].

*Paracoccidioides* spp. are also able to explore host proteins as iron sources, such as ferritin, transferrin, lactoferrin, and hemoglobin. However, hemoglobin available in the culture medium promotes the most robust growth of the fungus, compared to other host Fe proteins. This finding associated with the fact that *Paracoccidioides* spp. presents hemolytic potential and may have contact with hemoglobin/heme by hematogenous dissemination or when phagocytosed, inside macrophages, points to the fungal preference for hemoglobin/heme as an iron source [27,52,53]. In addition, the uptake of hemoglobin depends on a receptor-mediated mechanism, in a process similar to that described for *C. albicans* [11,27]. At the cytoplasmic proteomic level, fungal exposure to hemoglobin promotes the upregulation of enzymes related to the metabolism of amino acids, nitrogen, and sulfur and the downregulation of enzymes related to biosynthesis of porphyrin. This demonstrates that the fungus uses the molecule of hemoglobin not only as an iron source, but also as a source of nitrogen, sulfur and porphyrins [27]. Despite all of these findings, the changes on the surface of fungal cells triggered by the presence of hemoglobin had not been investigated yet.

The cell wall of *Paracoccidioides* spp. is the interface of contact between pathogen and the host. Therefore, the elucidation of the cell wall proteome is relevant for the identification of targets that are essential for the infectious process, which is preferably done under conditions that mimic the environment faced by the pathogen within the host. As an example, the study of the cell wall proteome of *C. albicans* after submission of the fungus to iron deprivation allowed the identification of several proteins related to Fe uptake from host sources, including the hemoglobin-receptors Rbt5 and Pga7, highlighting the importance of this type of approach [54]. Given the complexity of the heme/hemoglobin uptake event by *Paracoccidioides* spp., the objective of this study was to perform proteomic analyses of the pathogen’s cell wall, after treatment with hemoglobin and using Fe deprivation as control, in order to investigate other proteins, besides *Pb*Rbt5 that could contribute to this process.

Several proteins identified in the cell wall of *P. lutzii* are related to processes classically characterized at the cytoplasmic level (Table 1, Appendix A), such as ribosomal proteins, enolase, involved in glycolysis, as well histones related to DNA processing and nucleosome structure, among others. Although intriguing, our results are in accordance with findings in the literature. Approaches of the cell wall proteomes of *P. brasiliensis* and *C. albicans* also evidenced the presence of classic cytoplasmic proteins in the cell wall [28,55,56]. Some authors argue that the presence of these proteins in the cell wall could be due to the methodology used to obtain the protein extracts. Others argue that the occurrence of these proteins is due to the fact that they are bi or multifunctional (moonlighting) proteins, with different functions depending on the subcellular location there are expressed [57,58]. The process used in the present work to obtain cell wall proteins was based on extensive washes of the extracts with decreasing concentrations of NaCl solutions, in order to remove cytoplasmic contaminants associated to the cell wall by non-specific interactions [28,36].

In addition to the possible absence of cytoplasmic contaminants, the identified proteins were submitted to the analysis of prediction of secretion (by classical and/or non-classical route), which corroborated the proteins identified at the cell surface. Moreover, 39.8% of the proteins were predicted to be secreted via classical and/or non-classical pathways and they were grouped mainly as secreted by non-classical pathways (those that are independent of a signal peptide in the N-terminal portion of the protein), which are not fully understood yet. Several proteins might be secreted by mechanisms that still demand elucidation and consequently are not yet included in the search algorithms of the tools applied here [57,59].

Regarding the use of heme/hemoglobin by *P. lutzii*, *Pb*Rbt5 and other proteins related to iron uptake was not identified by the present proteomic approach [27]. This is due to the fact that Rbt5 is a receptor present in the cell wall retained by a remnant of GPI. To obtain cell wall samples enriched with anchored GPI proteins, other strategies are employed, such as treatment with hydrofluoric acid-pyridine [36]. Despite the absence of *Pb*Rbt5, other up-regulated proteins were predicted as potential adhesins (Table 1). The elongation factor-Tu (EF-Tu) was up-regulated (Table 1). Although it was not identified as an adhesin by the bioinformatics tools applied here, this protein was characterized as an adhesin in a previous work, contributing to the host–pathogen interaction [60]. Recent work performed by our group identified *P. lutzii* surface proteins that interact with macrophages [34]. We found that 25 out of 33 up-regulated proteins predicted to be secreted, were also identified as surface proteins interacting with macrophages by Tomazett et al. (2019) [34]. The upregulation of adhesins allows the inference that the pathogen treatment with hemoglobin mimics conditions found in the host. Thus, the fungus increases the expression of adhesins to improve interaction with the host and establish the infection [61]. We performed flow cytometry to confirm the increase in the adhesion capacity of *P. lutzii* in the presence of hemoglobin (Figure 2), which reiterates the importance of the hemoglobin molecule to the fungus.

We identified a differentially expressed HSP30 protein when the fungus is exposed to hemoglobin (Table 1 and Figure 1A). The HSP30’s orthologue in *P. brasiliensis* also presented positive regulation at the transcriptional level when the fungus was grown in the presence of hemoglobin (Figure 1B). Immunofluorescence analysis corroborated these findings, a strong fluorescence and recognition of HSP30 by polyclonal antibodies was observed at the cell surface of *P. lutzii* cells in the presence of hemoglobin (Figure 3). Interestingly, heme oxygenase proteins belong to the HSP30 family [62,63,64,65], a fact that deserves attention due to the cultivation condition employed in the present study. To investigate whether HSP30 is a heme oxygenase, we performed sequence and structural alignments with human heme oxygenase 1 (Appendix A). We found a level of similarity between these molecules, which suggests conserved functions. Despite a promising result, this in silico analysis require experimental validation.

We employed molecular dynamics analysis to determine the best conformation of interaction between HSP30 and hemoglobin. The pattern of interactions performed by heme-bound proteins influences several biological processes, such as signaling pathways and diseases. Several of those interactions have been identified over the past decades but their molecular basis and their consequences to infectious diseases such as PCM are poorly understood. In addition, the heme-regulatory motif is found within the structure of several proteins that interact with hemoglobin [66]. This motif contains a heme-coordination site and is placed on the surface of the protein facilitating its interaction with its partner [67]. The binding of heme groups to such motifs is able to change protein stability leading to a catalytically active heme-protein-protein complex. The histidine (H) and tyrosine (Y)-based motifs are the most prominent representatives for the interaction of heme-proteins [67,68]. Here, we found the residues Y222 interacting with F46 (phenylalanine), H45 with E230 (glutamic acid) and Y545 interacting with S339 (serine) (Figure 4E,F). Additionally, confirming the in silico findings, we showed that hemoglobin physically binds to the recombinant HSP30 by far-western (Figure 4G).

We investigated the importance of HSP30 in the context of hemoglobin utilization by *Paracoccidioides* spp. at the genetic level. The genetic manipulation of the *Paracoccidioides* genus is based on antisense RNA technology coupled with an ATMT [69]. This technology relies on targeted down-regulation of gene expression and allowed to point the relevance of the Rho-like GTPase PbCDC42 as a virulence determinant [70], to map SCONC (negative regulator of the inorganic sulfur assimilation pathway) of the sulfur metabolism [71] and the involvement of cytochrome c peroxidase to *P. brasiliensis* virulence [47] among others. In addition, this knock-down system evidenced the involvement of Rbt5 in the hemoglobin utilization by *Paracoccidioides* spp. [27] and the role of the biosynthesis of siderophores for the fungus virulence [48]. Although relatively recent, this technology has been of the current method of choice to manipulate *Paracoccidioides* spp. genetically [72,73,74,75,76,77,78].

Based on this methodology, we silenced *HSP30* in *P. brasiliensis* yeast cells. Under usual cultivation conditions, the silenced strain showed similar growth behavior and viability to the WT (Figure 5B). However, there was a 40% reduction in the transcriptional level of HSP30 in the silenced strain in the presence of hemoglobin, confirming the silencing event (Figure 5C). We observed that the AsHSP30 strain generated about 25% less CFU compared to WT in the presence of hemoglobin, a statistically significant value (*p* ≤ 0.05). This result highlights the importance of HSP30 in the context of hemoglobin utilization by *Paracoccidioides* spp.

Despite all of these findings, some questions still demand elucidation: is the interaction between hemoglobin and HSP30 transient? If HSP30 is a heme oxygenase, is the interaction of HSP30 with hemoglobin just one part of the iron uptake system? Noteworthy, the degradation of heme-by-heme oxygenases leads to the production of iron, biliverdin, and carbon monoxide in a reaction dependent on NADPH: cytochrome *p*-450 reductase. [79]. Does the iron released from hemoglobin by the action of HSP30 feed the fungal reductive iron uptake pathway or is captured by siderophores? Could HSP30 also have a cytoprotective function, as demonstrated for human heme oxygenase [80]? Interestingly, the heme group presents toxic potential because it is able to produce reactive oxygen species [81]. Cytochrome c peroxidase is related to the defense against oxidative stress [82] and was one of the upregulated proteins identified in the proteome in the presence of hemoglobin. Considering the toxic potential of the heme group, the upregulation of cytochrome c peroxidase and the possibility of HSP30 being a heme oxygenase, led us to infer that *Paracoccidioides* spp. employs mechanisms to counteract the toxicity caused by the heme group and thus enable the use of the molecule as a source of iron.

Additionally, if HSP30 acts as heme oxygenase, does its function dependent on an ancillary protein, and what protein would that be? Considering the production of carbon monoxide, could the HSP30 promote immunomodulatory effects as described for the *C. albicans* heme oxygenase [83]? It is notable that the present work opens perspectives for promising future investigations. Our results demonstrated that upregulation of potential adhesins occurs at the cell surface when the fungus is exposed to hemoglobin, which was confirmed by flow cytometry. We also confirmed the increased ability of the fungus to interact with macrophages. In addition, HSP30 of *Paracoccidioides* spp. is a novel hemoglobin-binding protein, silencing this gene decreases the ability of *P. brasiliensis* to use hemoglobin as a nutrient source. Henceforth, additional studies are needed to establish HSP30 as a virulence factor, which could support the development of new therapeutic and/or diagnostic approaches to PCM.

## Figures and Tables

**Figure 1 jof-07-00021-f001:**
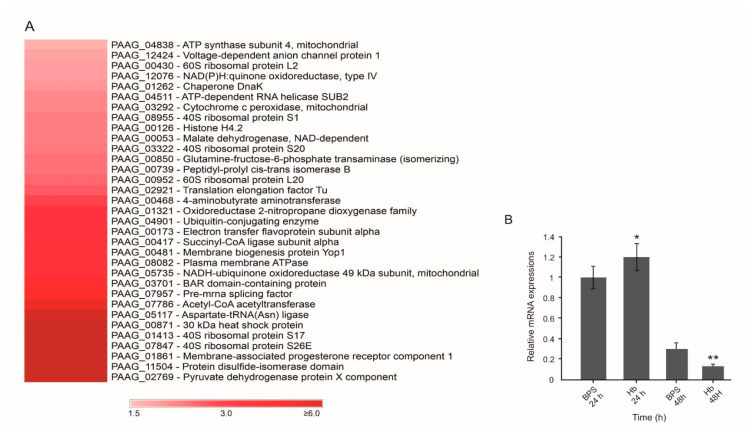
Heatmap of up-regulated proteins predicted as secreted and HSP30 transcriptional analysis. (**A**) The chart was generated using Microsoft Excel tools. Scale: ratio between Hb/BPS conditions. (**B**) HSP30 transcriptional analysis of WT strains of *P. brasiliensis* in the presence of hemoglobin. The yeast cells were grown in liquid Brain Heart Infusion (BHI) medium for 72 h and subsequently in McVeigh and Morton modified medium (MMcM) containing hemoglobin for 24 and 48 h. The control comprised cells grown in the presence of BPS under the same conditions. GraphPad Prism software (GraphPad Software, Inc., La Jolla, CA, USA) was used for the statistical analysis and Student’s *t*-test was applied considering *p* values ≤ 0.05 statistically significant. (*, **) denotes statistically significant differences.

**Figure 2 jof-07-00021-f002:**
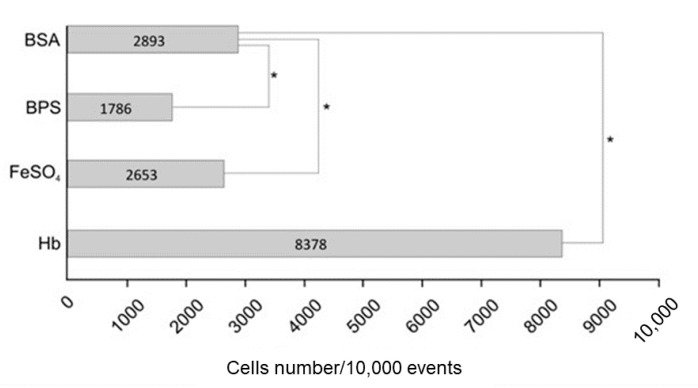
*P. lutzii* interaction with macrophages assessed by flow cytometry. *P. lutzii* yeast cells were grown in the presence of BPS, hemoglobin or FeSO_4_ and incubated for 4 h with RAW 264.7 macrophages. BSA was used as control. Methanol fixed, non-permeabilized cells were incubated with 100 µg/mL of Congo Red. The amount of *P. lutzii* cells that interacted with macrophages was assessed by flow cytometry using the instrument Guava^®^ easyCyte (MERK). Proportion test was used for statistical comparison between the conditions analyzed. (*) denotes a statistically significant difference.

**Figure 3 jof-07-00021-f003:**
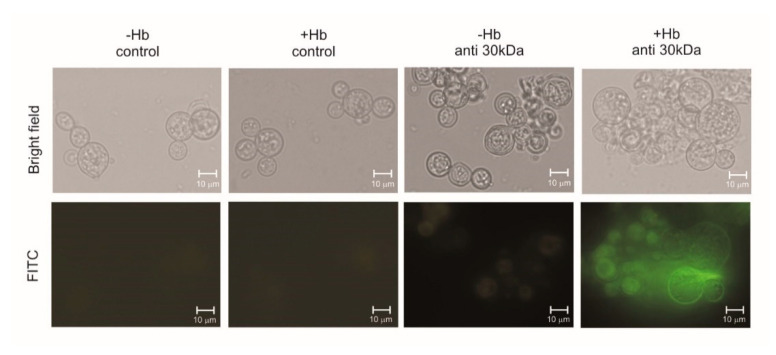
Analysis of hemoglobin influence on HSP30 protein expression at the cell surface of *P. lutzii*. Yeast cells were grown in the presence or absence of hemoglobin. Methanol fixed, non-permeabilized cells were incubated with anti-HSP30 antibodies or with pre immune serum for control. The pictures were taken in bright field and at 450/490 nm of fluorescein isothiocyanate (FITC) probe. All representative pictures were taken using an Axioscope microscope (Carl Zeiss AG, Berlin, Germany) and magnified 400×.

**Figure 4 jof-07-00021-f004:**
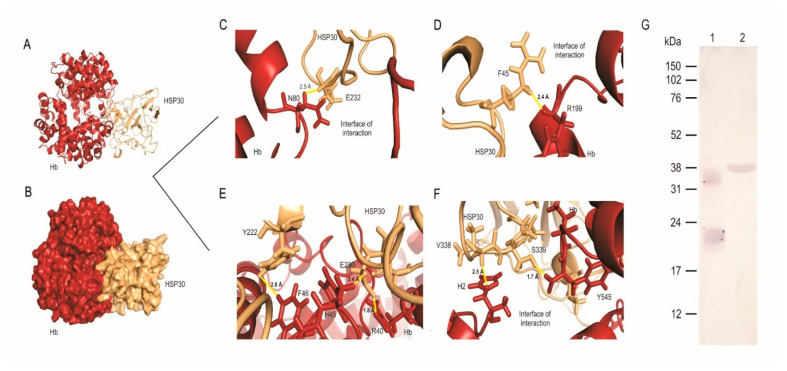
HSP30 is a hemoglobin-binding protein. (**A**) Cartoon view of the interaction showing the secondary structures of both proteins and the secondary structures that maintain the interface of interaction stable. (**B**) Surface view of the complex formed by HSP30 (beige) and hemoglobin (red). The interaction is maintained by a large interface of interaction between the proteins under study and by several amino acids within the interface of interaction. (**C**) Asparagine (N80) interacting with glutamic acid (E232) by a 2.5 Å hydrogen bond. (**D**) Arginine (R199) interacting with phenylalanine (F45) by a 2.4 Å hydrogen bond. (**E**) F46 interacts with tyrosine (Y222) and E230 through two hydrogen bonds, distant 2.8 and 2.4 Å, respectively. E230 also interacts with R40, distant 1.8 Å. (**F**) H2 (histidine) and Y545 interacts with V338 and S339, via hydrogen bonds distant 2.5 and 1.7 Å, respectively. (**G**) Far western analysis showing hemoglobin binding to PbHSP30. 1—Bovine hemoglobin recognition by anti-human hemoglobin monoclonal antibody. 2—Binding of hemoglobin to recombinant HSP30.

**Figure 5 jof-07-00021-f005:**
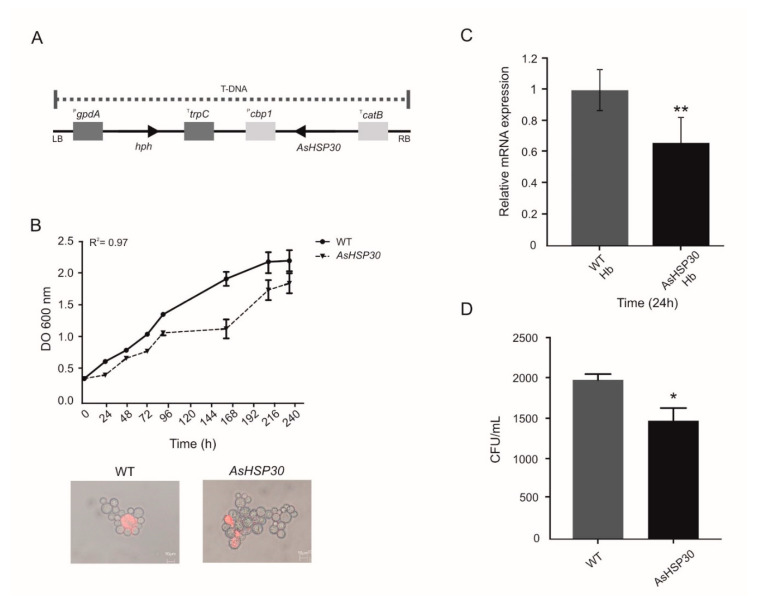
*HSP30* silencing in *P. brasiliensis* cells. (**A**) Integration DNA cassette scheme. The asRNA of *Pb*HSP30 was flanked by the calcium-binding protein promoter region (^P^*cbp1*) of *Histoplasma capsulatum* and by the cat-B termination region (^T^*catB*) of *Aspergillus fumigatus*. The hygromycin resistance gene used as a selection mark was flanked by the glyceraldehyde-3-phosphate dehydrogenase promoter region (^P^*gapdA*) and the trpC termination region (^T^*trpC*) of *Aspergillus nidulans*. (**B**) Growth and viability of *P. brasiliensis* strains. Yeast cells of WT and AsHSP30 were grown in liquid Brain Heart Infusion (BHI) medium for 10 days and the OD was measured daily at 600 nm. Cellular viability was examined by propidium iodide staining. (**C**) Transcriptional analysis of WT and AsHSP30 strain in the presence of hemoglobin. The strains were grown in liquid BHI medium for 72 h and subsequently in MMcM medium containing hemoglobin. The relative mRNA expression was evaluated at 24 h. (**D**) Number of WT and AsHSP30 CFUs recovered after growth in the presence of hemoglobin. After culturing in the presence of hemoglobin, WT and AsHSP30 cells were plated in solid BHI medium and the colony forming units were counted after 7 days. GraphPad Prism software (GraphPad Software, Inc., La Jolla, CA, USA) was used for statistical analysis and Student’s t-test was applied considering *p* values ≤ 0.05 statistically significant. (*, **) denotes statistically significant differences.

**Table 1 jof-07-00021-t001:** Up-regulated proteins in Fraction 1 (F1) of the cell wall predicted as secreted, following *Paracoccidioides lutzii* yeast cells exposition to hemoglobin and bathophenanthrolinedisulfonic acid (BPS) for 48 h.

Accession ^a^	Description ^b^	Score ^c^	Expression LevelsRatio (Hb/BPS) ^d^	SignalP ^e^	SecretomeP ^f^
PAAG_00871	30 kDa heat shock protein (HSP30) •	754.8	*	-	0.786
PAAG_08955	40S ribosomal protein S1	672.2	1.62	-	0.720
PAAG_01413	40S ribosomal protein S17	1850.1	*	-	0.738
PAAG_03322	40S ribosomal protein S20	1303.5	1.63	-	0.750
PAAG_07847	40S ribosomal protein S26E	409.6	*	-	0.613
PAAG_00468	4-aminobutyrate aminotransferase	977.2	1.75	-	0.601
PAAG_00430	60S ribosomal protein L2	360.1	1.55	-	0.853
PAAG_00952	60S ribosomal protein L20	1022.6	1.67	-	0.712
PAAG_07786	Acetyl-CoA acetyltransferase	1043.1	4.10	-	0.655
PAAG_05117	Aspartate-tRNA (Asn) ligase	565.1	5.53	-	0.609
PAAG_04838	ATP synthase subunit 4, mitochondrial	663.4	1.51	-	0.781
PAAG_04511	ATP-dependent RNA helicase SUB2	2259.1	1.60	-	0.722
PAAG_03701	BAR domain-containing protein	844.8	2.48	-	0.614
PAAG_01262	Chaperone DnaK	2982.3	1.57	0.864	-
PAAG_03292	Cytochrome c peroxidase, mitochondrial •	3045.4	1.60	-	0.809
PAAG_00173	Electron transfer flavoprotein subunit alpha	465.5	1.88	-	0.642
PAAG_00850	Glutamine-fructose-6-phosphate transaminase (isomerizing)	1305.8	1.65	-	0.693
PAAG_00126	Histone H4.2	4337.0	1.62	0.792	-
PAAG_00053	Malate dehydrogenase, NAD-dependent	1037.2	1.62	-	0.651
PAAG_00481	Membrane biogenesis protein Yop1	922.3	1.93	-	0.902
PAAG_01861	Membrane-associated progesterone receptor component 1	443.0	*	-	0.735
PAAG_12076	NAD(P)H:quinone oxidoreductase, type IV •	1770.3	1.55	0.718	-
PAAG_05735	NADH-ubiquinone oxidoreductase 49 kDa subunit, mitochondrial	668.4	2.05	-	0.675
PAAG_01321	Oxidoreductase 2-nitropropane dioxygenase family •	2254.9	1.79	-	0.707
PAAG_00739	Peptidyl-prolyl cis-trans isomerase B	583.7	1.65	0.641	-
PAAG_08082	Plasma membrane ATPase	738.9	1.93	-	0.712
PAAG_07957	Pre-mRNA splicing factor •	539.9	2.69	-	0.801
PAAG_11504	Protein disulfide-isomerase domain •	364.5	*	-	0.783
PAAG_02769	Pyruvate dehydrogenase protein X component •	325.5	*	-	0.685
PAAG_00417	Succinyl-CoA ligase subunit alpha	1400.3	1.88	-	0.624
PAAG_02921	Translation elongation factor Tu	990.0	1.70	-	0.773
PAAG_04901	Ubiquitin-conjugating enzyme	555.9	1.79	-	0.883
PAAG_12424	Voltage-dependent anion channel protein 1	2827.7	1.54	-	0.761

^a^ Protein accession number in NCBI, available at https://www.ncbi.nlm.nih.gov/protein. ^b^ Description of the protein in the *Paracoccidioides* spp. databank, available in https://www.uniprot.org/uniprot/?query=paracoccidioides+lutzii&sort=score. ^c^ Score of the quality of protein identification. ^d^ Ratio between quantification of proteins identified in hemoglobin/iron deprivation. Values ≥ 1.5 indicate upregulated proteins; (*) indicates that the protein was identified only upon hemoglobin treatment. ^e^ Prediction of signal peptide presence; score ≥ 0.45; prediction performed by SignalP 4.1 available at http://www.cbs.dtu.dk/services/SignalP/. (-) indicates that the signal peptide was not identified. ^f^ Prediction of protein secretion by non-classical pathways; score 0.6; prediction performed by SecretomeP 2.0 available at http://www.cbs.dtu.dk/services/SecretomeP/. (-) indicates that the protein was not predicted as secreted by non-classical pathways. ^•^ Prediction of adhesin function; score ≥−0.8; predicted by the FaaPred tool available at http://bioinfo.icgeb.res.in/faap/.

## Data Availability

Not applicable.

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
