# Peer review of "Interacting with Hemoglobin: Paracoccidioides spp. Recruits hsp30 on Its Cell Surface for Enhanced Ability to Use This Iron Source"

_jof, 2021, doi:10.3390/jof7010021_

Round 1

Reviewer 1 Report

This manuscript by de Souza, et al., describes the identification of heat-shock protein 30 (HSP30) for its role in utilization of hemoglobin as an iron source in Paracoccidioides spp. Authors identified multiple proteins in the cell wall fraction from the cells grown in the presence of hemoglobin using nano UPLC-mass spectrometry. Study shows that under the presence of hemoglobin as an iron source, level of HSP30 increases at the cell surface that binds to the hemoglobin and involves in utilization of this iron-source. Authors used. in-silico and molecular simulation analysis to study the interaction between HSP30 and hemoglobin.

My specific comments are following

1. I am surprised to see a significant reduction in the expression of HSP30 mRNA at 48 h in comparison to 24 h. This points to a fact that the role of HSP30 might be only relevant at the early growth stages of the fungus, however Figure 5b, demonstrate that growth of the fungus upon silencing of HSP30 is not significantly different than WT at early time-points  in hemoglobin containing-medium. Does it mean that the expression of HSP30 is just a general stress-response (nutrient-starvation) of the fungus. Does the silencing of HSP30 causes a general growth defect of the fungus? Or if the silencing observed in the strain was not enough? Does it also mean that some other HSP30-independent mechanisms for hemoglobin utilization are present and are more important?

2. Authors haven’t been able to systematically differentiate the functional relevance of HSP30 either as a hemoglobin binding protein or an adhesin protein. A clearer approach will be to carry out a detail general growth analysis and macrophage adhesion assays with the HSP30-silenced strain of the fungus and hemoglobin binding assays. Authors can utilize microscopic and biochemical approaches to provide a clear answer about the actual role of HSP30.

3. Considering authors already have a HSP30 silenced strain in their hand, it is appropriate to do functional characterization related to its importance in vivo survival inside macrophages or mice.

4. Figure 2 requires statistical representation and clear idea about the number of independent experiments performed.

5. In Figure 3, how do authors know if the anti-HSP30 antibody is not binding to any cell surface bound hemoglobin in microscopic panel 4. Cell fixation will also fix any cell-surface bound and uninternalized hemoglobin. Currently this doesn’t explain whether the hemoglobin presence increases protein expression of HSP30 or it’s just the binding of HSP30 to hemoglobin.

6. Authors should provide a biochemical hemolytic activity for the role of HSP30 as a heme oxygenase or should avoid highlighting it majorly.

Minor comments:

  1. Line 56-57, A recent work by Bairwa et al., Cell Microbiol. 2019, also describes important role of clathrin-mediated endocytosis in hemoglobin use in neoformans.
  2. Line 362: Rephrase to ‘proteins were selected’

Reviewer 2 Report

The manuscript by Ferreira de Souza and colleagues describe the characterization of HSP30 of P. lutzii as a novel hemoglobin-binding protein. P. lutzii is a dimorphic fungus that is one of the causal agents of paracoccidioidomycosis. When deprived of micronutrients, P. lutzii responds by utilizing hemoglobin as a source of iron. While the GPI-anchored receptor Rbt5 is generally utilized as the primary mechanism for acquiring iron from hemoglobin, the authors investigated the possibility that other proteins may also play a role in iron acquisition.

Employing various biochemical and molecular biology techniques, the authors set out to identify and characterize proteins in P. lutzii that were regulated in the presence of hemoglobin as the only iron source and were able to identify HSP30 as a potential hemoglobin-binding protein. 

Major points:

  • Do not use abbreviations in the abstract.
  • Define the abbreviation in its first use and then abbreviate thereafter.
  • The material and methods and results sections of the manuscript require language editing. The discussion requires minor language editing.
  • The authors must discuss why Rbt5 was not identified as 'hit' in their discovery.
  • Table 1 - the authors state "Only proteins with satisfactory score values were included in the analyzes". The authors must quantify this statement. What is their definition of 'satisfactory score values'? Were these values benchmarked against other similar publications?
  • The following comment in Figure 1 is unclear - "(*) indicates that the protein was identified only upon hemoglobin treatment". The title states "Proteins increased in Fraction 1 (F1) of the cell wall predicted as secreted, after Paracoccidioides lutzii yeast cells exposition to hemoglobin for 48
    hours", indicating that all the proteins in the table were identified following hemoglobin treatment.

Reviewer 3 Report

This is an interesting study providing some evidence in favor of a hypothesized mechanism of host pathogen interaction in paracoccidiodomycosis. The authors have approached the investigation from multiple angles, which are noteworthy. However, I am concerned about some lacunae in the overall presentation. The following are my suggestions, which I hope would be helpful to the authors for a revision.

Line 22: change Fe to iron for consistency.

Line 24: Please rewrite this sentence to clearly specify Rbt5 is a fungal receptor.

Lines 26-28: "we coupled... iron source." Please rewrite this sentence. The current placement of phrases as well as of multiple commas is making it difficult to understand.

Lines 28-29: "Proteins from ... of iron." Please rewrite this sentence to correct grammatical errors. As currently written, it is difficult to understand which cell wall is the source of F1 and which cells are deprived of iron. When you write "in comparison to", there must be a comparison between similar items. The current structure compares proteins with cells. Please correct.

Line 30: "hemoglobin exposure... macrophages." Please eliminate this line or rewrite it with more details. As currently written, it is not clear at all why Hemoglobin exposure would increase fungal adherence to macrophages (which have nothing to do with hemoglobin).

Line 33: ATMT, if this is not a standard abbreviation for the journal, please define on first use.

Line 48: Change Fe to iron. Sudden inclusion of the symbol here is not consistent with the rest of the text.

Line 50 and paragraph. "Iron Source". In defining hemoglobin as the iron source, this paragraph is missing the information on how (a) heme is separated by fungal cells from host hemoglobin, and (b) iron, which is bound to the host heme, is actually detached and transferred to the fungal cell. Neither of the examples provided (Candida and Cryptococcus) mentions this. Please add a few lines elucidating this.

(You do mention this in the context of paracocci, in lines 70-72. But this mechanism is important to the understanding of the process and should be mentioned in the previous paragraph.)

Line 82: place the comma BEFORE thereby, not after.

Lines 91-92: This is the same exact sentence as in the Abstract and has the same exact problem: it is not clear. Please rephrase.

Lines 94-95: "Cell wall's proteins... of iron." This sentence reflects the sentence in the Abstract and is equally unclear. Please rephrase.

Line 117 and rest of section 2.2: (NOT A CORRECTION, just CLARIFICATION) If I am reading your method correctly, the paracocci yeasts were grown in BHI, then transferred to MMcM+BPS (to completely deplete intracellular Fe), THEN divided into two groups: 1) placed in MMcM+Hb (test group) and 2) placed in MMcM+BPS (Control group).Am I understanding correctly? Is there any viability loss due to the first incubation in MMcM+BPS?

Line 130: "buffer added of glass beads" is unclear. Please rephrase.

Line 131: (QUESTION) Do glass beads not precipitate with the cell pellet at 800 x g? How long is the spin? Please indicate the time in the text.

Line 132: (QUESTION) What is the rationale behind this decreasing concentration of NaCl? (You write this only in the discussion; I suggest you mention this in the Methods as well.) Did you encounter any material loss during these multiple rounds of wash steps?

Line 154: define PHB on first use.

Section 2.7. After the NanoUPLC-MS described in the previous section, section 2.7 abruptly starts talking about HSP30. Please provide a bridging statement (single sentence is fine) to connect the previous section to the focus on HSP30 in the current one.

Line 221: What kind of mice for the Ab production? How many mice were used?

Line 234: mention the host species of the primary MAb.

Line 375: Was this decrease in expression level related to loss of cell viability due to Fe-specific phenomena or was it just natural decrease?

Line 95 of Page 16: "similar growth and viability" is not strictly correct based on the Figure B. How did you normalize the inputs in each group for Figure 5C and 5D?

Line 120 of Page 17: what is meant by "host's nutritional immunity"?

Line 123, 127, 137, 144 of Page 17: Please change Fe to iron.

I would also suggest trimming your conclusions to adhere to the exact data you have obtained; please take another look at some of the conclusions using the HSP30 silencing data.

Also, please have your manuscript carefully edited for the language use.

Round 2

Reviewer 1 Report

Authors have appropriately addressed and answered my questions. Thanks.